# A Current State of Proteomics in Adult and Pediatric Inflammatory Bowel Diseases: A Systematic Search and Review

**DOI:** 10.3390/ijms24119386

**Published:** 2023-05-27

**Authors:** Ondrej Fabian, Lukas Bajer, Pavel Drastich, Karel Harant, Eva Sticova, Nikola Daskova, Istvan Modos, Filip Tichanek, Monika Cahova

**Affiliations:** 1Clinical and Transplant Pathology Centre, Institute for Clinical and Experimental Medicine, 140 21 Prague, Czech Republic; eva.sticova@ikem.cz; 2Department of Pathology and Molecular Medicine, 3rd Faculty of Medicine, Charles University and Thomayer Hospital, 140 59 Prague, Czech Republic; 3Department of Gastroenterology and Hepatology, Institute for Clinical and Experimental Medicine, 140 21 Prague, Czech Republic; lukas.bajer@ikem.cz (L.B.); pavel.drastich@ikem.cz (P.D.); 4Institute of Microbiology, Czech Academy of Sciences, 142 20 Prague, Czech Republic; 5Proteomics Core Facility, Faculty of Science, Charles University, 252 50 Vestec, Czech Republic; karel.harant@natur.cuni.cz; 6Department of Pathology, Royal Vinohrady Teaching Hospital, Srobarova 1150/50, 100 00 Prague, Czech Republic; 7Experimental Medicine Centre, Institute for Clinical and Experimental Medicine, 140 21 Prague, Czech Republic; nikola.daskova@ikem.cz (N.D.); monika.cahova@ikem.cz (M.C.); 8Department of Informatics, Institute for Clinical and Experimental Medicine, 140 21 Prague, Czech Republic; istvan.modos@ikem.cz (I.M.); filip.tichanek@ikem.cz (F.T.)

**Keywords:** Crohn’s disease, inflammatory bowel disease, pediatric, proteomics, proteome, ulcerative colitis

## Abstract

Inflammatory bowel diseases (IBD) are systemic immune-mediated conditions with predilection for the gastrointestinal tract and include Crohn’s disease and ulcerative colitis. Despite the advances in the fields of basic and applied research, the etiopathogenesis remains largely unknown. As a result, only one third of the patients achieve endoscopic remission. A substantial portion of the patients also develop severe clinical complications or neoplasia. The need for novel biomarkers that can enhance diagnostic accuracy, more precisely reflect disease activity, and predict a complicated disease course, thus, remains high. Genomic and transcriptomic studies contributed substantially to our understanding of the immunopathological pathways involved in disease initiation and progression. However, eventual genomic alterations do not necessarily translate into the final clinical picture. Proteomics may represent a missing link between the genome, transcriptome, and phenotypical presentation of the disease. Based on the analysis of a large spectrum of proteins in tissues, it seems to be a promising method for the identification of new biomarkers. This systematic search and review summarize the current state of proteomics in human IBD. It comments on the utility of proteomics in research, describes the basic proteomic techniques, and provides an up-to-date overview of available studies in both adult and pediatric IBD.

## 1. Introduction

Inflammatory bowel diseases (IBD) represent chronic systemic immune-mediated conditions with a predilection for the gastrointestinal (GI) tract and include Crohn’s disease (CD) and ulcerative colitis (UC) as two major phenotypes. Currently, IBD are perceived more likely as a continuous spectrum of disorders ranging from CD with a small bowel and/or upper GI predilection across colonic CD to IBD unclassified (IBDU) to distal UC [1,2]. The annual incidence of CD ranges between 0.3 and 12.7/100,000 persons and 0 and 20.2/100,000 in Europe and North America, respectively. For UC, the annual incidence is 0.6–24.3/100,000 in Europe and 0–19.2/100,000 in North America. The highest incidence worldwide can be seen in the Faroe Islands, reaching up to 81/100,000 inhabitants [3,4]. There is a well-known west–east and north–south descending gradient with the highest occurrence of the disease in countries in Northern and Western Europe and North America [5]. Although the disease may manifest at any age, young adults are the most commonly affected group. The second peak can be seen in older patients around their seventh decade. Pediatric IBD represent approximately 1/4 of all cases [6,7]. The etiology of the disease remains largely unknown, and the pathogenesis is poorly understood. However, the impaired homeostasis between the intestinal epithelial barrier, immune barrier, and commensal intestinal microbiota in a genetically susceptible host is considered to be a key mechanism of the disease’s initiation and progression (Figure 1) [8].

The advances in the diagnostics and implementation of modern therapeutic approaches improved the overall clinical management of patients. However, up to 6% of adult and 13% of pediatric IBD cases remain further unclassified and many patients still face an adverse clinical course with frequent recurrences or severe complications [7]. The variable clinical presentation and uneven response to therapy suggest that etiopathogenic mechanisms of the disease progression are probably prone to substantial inter-individual variability. There is, thus, demand for new invasive or non-invasive biomarkers that would facilitate a diagnostic process, reflecting the actual intensity of inflammation and predicting the subsequent clinical course or development of complications.

A biomarker is defined as a measurable indicator of a specific biological state [9]. Many promising serum, fecal, or tissue biomarkers were introduced over time, but only a small fraction was implemented in routine clinical practice. The ideal biomarker should be non-invasive, highly sensitive and specific, easily detectable, and financially reasonable. To this day, no such biomarker has been identified. Proteomics represents a promising method of identification of potential new biomarkers based on the analysis of a large spectrum of proteins from tissue samples or body fluids. This systematic search and review summarizes the up-to-date proteomic research in adult and pediatric IBD, briefly describes basic proteomic techniques, comments on the value of proteomics in biomarker identification, and provides a list of available studies.

### 1.1. The Value of Proteomics

The term proteome describes a full set of proteins translated from a transcriptome within a single cell, tissue, or specific cellular compartment [10,11]. Proteomics then refers to a large-scale study analyzing a proteome. In practice, it is understood as a research sphere aiming at the identification of the largest number of proteins or peptides possible in the given tissue, clarifying their functions and understanding their reciprocal interactions [12]. It is well known that the number of specific protein variants in our tissues greatly exceeds the number of protein-coding genes [13]. Currently, more than 20,000 such genes are known. Each gene may give rise to several different RNA transcripts, which are subsequently translated to proteins. The proteins themselves may undergo a series of post-translational modifications, such as phosphorylation, glycosylation, or methylation. Currently, several such modifications are recognized. Each protein, thus, may give rise to hundreds of specific proteoforms [13,14]. Many additional arrangements throughout the transcriptomic or translational process also play a role, such as the regulation of translational rate or diversity of protein transport [13,14,15]. It is estimated that hundreds of thousands or maybe even millions of specific proteoforms exist in the human body [14,16]. It is, thus, obvious that a spectrum of potential proteins involved in the specific biological process cannot be reliably estimated solely via genomic or transcriptomic analysis.

In IBD, a genetic background plays an important role in the etiopathogenesis of the disease. Advances in molecular genetics and the introduction of genome-wide association studies allowed the detection of many candidate genes, whose pathogenic mutations or single nucleotide polymorphisms show an association with an increased risk of CD or UC. However, it is the proteomics that may finally link genomic and transcriptomic alterations with the phenotypical presentation of IBD and allow an understanding of mechanisms responsible for the initiation and progression of the disease. Previously, a search for candidate biomarkers was limited to a shortlist of proteins suspected to be involved in IBD pathogenesis. Advancements in modern proteomic techniques, such as mass spectrometry (MS) and implementation of complex bioinformatic analyses, enabled the isolation and analysis of a large number of proteins, increasing the chance of a novel biomarker discovery [17].

Standard proteomic research aiming at the identification of new potential biomarkers follows several phases, including biomarker discovery, verification, and validation [18]. The first phase begins with an analysis of a tissue, body fluid (i.e., serum), stool, or isolated cells, from which a large spectrum of proteins is identified. Those subjects with the biggest difference in expression between the cohorts are further examined and several candidate proteins are proposed. Since this phase is usually both financially and time demanding, studies are often performed on small cohorts of patients with a high risk of false negative or false positive findings [9,18]. The subsequent verification phase tests the sensitivity and specificity of the candidate biomarkers on larger cohorts of patients. Usually, antibody-based techniques, such as enzyme-linked immunosorbent assay (ELISA) or Western blot, are applied. The amount of candidate biomarkers is usually significantly reduced after this phase [9,17,18].

The newly described biomarkers can serve several clinical purposes. In the context of IBD, they can facilitate the following actions: (1) the distinguishing of IBD from other diseases; (2) the differentiation of IBD phenotypes; (3) the understanding of the pathogenic processes in the respective IBD subtypes; (4) the monitoring of the disease activity; (5) the prediction of severe clinical complications or a disease relapse; (6) the prediction of responsiveness to therapy, and (7) the prediction of neoplastic transformation.

### 1.2. Techniques in Proteomics

Proteomic studies usually follow one of the two main analytical approaches termed “top-down” and “bottom-up”. The bottom-up approach identifies proteins based on the analysis of small peptide fragments created via previous enzymatic proteolysis [19]. The top-down strategy, on the other hand, recognizes whole proteins that did not undergo previous enzymatic lysis [20]. The first approach is a commonly used strategy of protein identification but entails a risk of losing less abundant protein fragments, a higher percentage of peptide overlaps, and an inferior quality of post-translational modification analysis. The second approach has a lower sensitivity in the large-scale identification of proteins from complex mixtures and is prone to a higher dynamic range challenge in protein identification and characterization. However, it may be better suited for the analysis of proteins with multiple different proteoforms [21,22].

The most commonly used proteomic techniques can be subclassified into two main categories, termed “gel-based” and “gel-free”. Gel-based techniques include polyacrylamide gel electrophoresis (PAGE), two-dimensional difference gel electrophoresis (2-DE of 2D-PAGE), and difference gel electrophoresis (DIGE). These techniques separate proteins on the basis of their isoelectric point and, in the case of the two-dimensional approach, also based on the molecular mass. The individual protein groups are then represented by spots in the gel medium [23,24]. The spots subsequently undergo an in-gel digestion via endopeptidases or enzymatic digestion after their excision from the gel. As the next step, they are usually analyzed via MS [25]. However, gel-based methods are labor-intensive and limited in their ability to detect low-abundance proteins or proteins with a high molecular mass [26]. Gel-free methods are much more straightforward. Sample is homogenized and directly proteolytically cleaved. Resulting peptides are separated usually via reverse phase liquid chromatography (LC) and eluting peptides are detected with MS. LC is a technique able to separate peptides from digested cell lysates for a subsequent MS analysis on the basis of their chemical properties, mainly on hydrophobicity. The analyzed molecules interact with stationary phase attached to the surface of sorbent beads in LC column. Molecules are separated while moving through column with the aid of a mobile phase [27,28]. The most commonly used approach for peptide quantification is a label-free approach in data dependent (DDA) or data independent (DIA) acquisition mode. In DDA mode, each fragmentation spectrum originates from the fragmentation of one isolated precursor. On the other hand, in DIA mode, multiple precursors are fragmented together, and the resulting fragments are assigned to peptide precursors during data processing [29,30]. Alternatively, isobaric tags for relative quantification (iTRAQ) or stable isotope labelling with amino acids in cell culture (SILAC) can be used. iTRAQ is a method based on the covalent binding of the N-terminus and side chain amines of peptides with isobaric tags of varying mass [31]. A similar approach to iTRAQ is a tandem mass tag (TMT) that varies with regard to the chemical structure of the used tags [32]. SILAC is a method in which stable isotope-labeled amino acids are incorporated into the proteome. It is usually performed on cell mixtures containing two cell populations. One population is fed with a growth medium containing labeled amino acids. Proteins from both cell populations are then combined and analyzed together via a MS. The labeled and non-labeled amino acids have a known mass shift that allows the comparison of a given peptide between both samples. The ratio of peak intensities in the mass spectrum for such peptide pairs reflects the abundance ratio for the two proteins [33,34,35].

Regardless of the protein preparation technique, the process is usually followed by a MS analysis. The basic workflow can be summarized in three steps: (1) protein/peptide ionization, (2) separation of ions on the basis of their mass-to-charge ratio, and (3) ion detection. If the analysis follows a top-down approach, the samples are usually analyzed directly via a MS. In bottom-up strategy, the proteins are cleaved to peptides and consequently separated via LC or directly analyzed via a MS. The most common ionization sources include matrix-assisted laser desorption/ionization (MALDI) and electrospray ionization (ESI). MALDI technique is usually preferred due to its capacity to produce singly charged ions of peptides, thus minimizing the complexity of the given spectra. The most frequently used ion analyzers are time-of-flight (TOF) or Fourier transform-based instruments (Orbitrap, FT-ICR-MS). In most instances, peptides undergo several rounds of fragmentation and MS analysis, termed tandem MS or MS/MS. In modern proteomics, ESI-based hybrid instruments are usually combined quadrupole with a TOF- or Orbitrap-based spectrometer. This technique is used to determine peptide sequences, while standard MS workflow is usually reserved for peptide mass measurements [36,37,38]. A flowchart showing the basic proteomic approaches is shown in Figure 2.

## 2. Methods

This systematic search and review was developed in accordance with the PRISMA reporting guidelines [39]. A comprehensive search of the MEDLINE database using the PubMed interface (http://pubmed.ncbi.nlm.gov [accessed on 8 January 2023]) was performed between 2–8 January 2023 to incorporate all IBD studies using high-throughput proteomic techniques published between January 2000 and January 2023. The search criteria included the following keywords: ((“proteomic”) OR (“proteomics”) OR (“spectrometry”) OR (“MALDI”) OR (“protein assay”)) AND ((“IBD”) OR (“Crohn’s”) OR (“ulcerative colitis”) OR (“inflammatory bowel disease”) OR (“inflammatory bowel diseases”)). The search was limited to studies performed on humans and written in English. Studies on animal models, plants, or cell cultures and review articles with previously published cases or editorials were excluded from the search. After the initial search strategy, all full-text articles were read through in detail and the studies that did not fulfill the inclusion criteria were excluded. Simultaneously, the reference lists of all articles were searched for further eligible articles. The final list of studies was included in this review. A flowchart showing the search process is displayed in Figure 3.

## 3. Results and Discussion

The preliminary search from the MEDLINE database found 144 records. Following the initial screening and exclusion of non-eligible records, 89 full-text articles proceeded to manual assessment. After that, 37 articles were definitely considered eligible. The subsequent reference lists search found an additional 25 records. In the end, 62 original studies [40,41,42,43,44,45,46,47,48,49,50,51,52,53,54,55,56,57,58,59,60,61,62,63,64,65,66,67,68,69,70,71,72,73,74,75,76,77,78,79,80,81,82,83,84,85,86,87,88,89,90,91,92,93,94,95,96,97,98,99,100,101] were included in the final review. Due to the large number of relevant articles found, a further selection of the studies that will be discussed in greater details was made. The selection process was based on multiple criteria, including the recency of the study, the size of the analyzed cohorts, the clinical importance of the main findings and recognition of the respective studies in subsequent works and reviews, ensuring that the most relevant articles for each topic will be addressed. These studies, their findings, and applied proteomic techniques are provided in Table 1. Table 2 summarizes discovered candidate biomarkers. A Appendix A provides a list of main statistical methods used for data analyses in the respective studies. A comprehensive list of all reviewed studies is provided in Appendix A.

### 3.1. Proteomics for Distinguishing IBD Patients from Other Intestinal Diseases or Healthy Controls

As mentioned earlier, the etiopathogenesis of IBD is still not fully elucidated. The diagnostic process represents a complex algorithm based on a combination of several diagnostic modalities, including endoscopy, radiology, histopathology, or even genetics. Moreover, many other infectious and non-infectious diseases may closely resemble IBD on both clinical and morphological levels. The proper diagnosis of IBD, thus, remains challenging.

In 2007, Meuwis MA et al. [40] performed a proteomic analysis of sera obtained from 30 patients with CD, 30 patients with UC, 30 patients with other intestinal inflammatory disorders, and 30 healthy controls (HC). Several proteins differentially expressed in the IBD group were identified, including platelet factor 4, haptoglobin α2, fibrinogen-α chain, and myeloid-related protein 8. These acute phase reactant proteins are produced by phagocytes, megakaryocytes, and other cells at the site of acute injury and may hold promise as potential biomarkers for IBD.

Shkoda A et al. [42] analyzed intestinal epithelial cells obtained from six patients with CD and six patients with UC. Another six patients with colorectal cancer (CRC) represented a control group. Panels of proteins upregulated in IBD were identified, including L-lactate dehydrogenase, carbonyl reductase, and keratin 19 a Rho-GDP. According to the authors, the proteins are mostly involved in signal transduction, stress response, and energy metabolism. The authors also correlated protein spectra from the inflamed and non-inflamed mucosal regions of the same patients and identified up to 40 proteins showing a different pattern of expression. The most differentiating proteins were programmed cell death protein 8 and annexin A2, which is an RNA-binding protein involved in the regulation of cell growth [102].

Moriggi M et al. [59] studied both intestinal tissue samples and isolated intestinal epithelia in a cohort of 30 CD patients and 30 UC patients. The aim was to investigate proteomic profiles in inflamed and non-inflamed mucosal regions and correlate them with HC (n = 16). The authors identified several differentially expressed proteins between inflamed and normal regions of mucosa in the IBD group, mostly involved in cytoskeleton rearrangement, cellular metabolism, autophagy, and extracellular matrix composition. Non-inflamed mucosa from UC patients showed alterations in keratins, vimentin, focal adhesion kinase, vinculin, and detyrosinated α-tubulin. On the other hand, inflamed UC regions showed changes in the expression of collagen type VI alpha 1 chain, tenascin C and vimentin. Inflamed mucosal regions in CD patients demonstrated upregulation of collagen type VI alpha 1 chain and vimentin, whereas normal-appearing mucosa had alterations in tenascin C, detyrosinated α-tubulin, vinculin, FAK, RHO-associated protein kinase 1, and vimentin.

Another study by Ning L et al. [60] on a cohort of 18 IBD patients (9 CD, 9 UC) and 6 HC patients identified four proteins associated with IBD, including upregulated CD38, chitinase 3-like 1, olfactomedin 4, and downregulated intelectin 1. The proteins are involved in the nicotinamide–adenine–dinucleotide metabolism and several signaling pathways promoting the intestinal inflammation [57,103,104]. CD38 protein was subsequently validated on an independent cohort of CD, UC, and HC patients (three in each group) using immunohistochemistry (IHC), immunofluorescence, and Western blot.

The study by Han NY et al. [51] performed on a small cohort of three patients with CD, four patients with UC, two patients with inflammatory polyps in the terrain of UC, and 3 patients with HC showed alteration in several proteins involved in a blood coagulation pathway, which were overexpressed in the IBD group. Specifically, patients with active UC showed upregulation of tau tubulin kinase 2, spectrin repeat containing nuclear envelope 2, succinate CoA ligase 2, GDP-forming subunit, and periostin. Patients with an active CD had increased expression of bone marrow proteoglycan, L-plastin, and proteasome activator subunit 1. Both CD and UC patients had higher levels of fibrinogen, calprotectin, myeloperoxidase, and neutrophil defensin-1 compared to the HC.

### 3.2. Proteomics for Differentiating CD from UC

Both CD and UC may manifest atypical morphology and ill-defined clinical symptoms, rendering a quick diagnosis followed by tailored therapy difficult. As a result, the percentage of patients diagnosed with IBDU or wrongly classified into the CD or UC categories is still relatively high [7]. This issue may yield significant consequences regarding distinctive induction and maintenance therapy. For instance, patients with CD are usually initiated via corticosteroids or exclusive enteral nutrition [105,106], whereas UC patients are treated using local or systemic 5-aminosalicylic acid [107]. Proper classification of IBD patients is, thus, essential.

M’Koma AE et al. [45] compared inflamed and non-inflamed mucosal and submucosal regions obtained from 27 patients with UC and 24 patients with colonic CD. Promising results showed the analysis of submucosal samples, where the comparison of inflamed regions of CD and UC patients led to the identification of two significant mass/charge number of ions (m/z) peaks (namely 8733 and 9245), while analysis of non-inflamed regions identified another three m/z peaks (31,752, 4939 and 5677). As a result, three peaks (2778, 9232 and 9519) allowed differentiation between CD and UC with 70% accuracy. However, specific proteins representing the spectral peaks need to be identified in subsequent studies.

In the study by Seeley EH et al. [49], the authors compared 26 samples from patients with colonic CD and UC. Data obtained from inflamed mucosal regions were used to train a support vector machine algorithm to discriminate between CD and UC. Based on the analysis of 25 spectral peaks, the algorithm was able to discriminate between colonic CD and UC with 76.9% accuracy. Similar to the previous study, specific proteins behind the spectral peaks are yet to be elucidated.

Arafah K et al. [64] examined protein spectra obtained from nine patients with CD and nine patients with UC. Several proteins upregulated in CD were identified, including keratin 4, neutrophil elastase, lactotransferrin, lysozyme C, protein S100 A8 and A9, cathepsin G, the protein group Ig mu chain C region, and Ig mu heavy chain disease protein. These results suggest a different neutrophilic activity between CD and UC and a different implication of damage-associated molecular patterns between the two diseases [108,109,110]. Apart from that, the aldo-keto reductase family 1 member C3 protein was expressed in 8/9 CD patients and was absent in the UC group. The presence of the protein was subsequently confirmed via IHC on the validation cohort of 62 CD patients and 51 UC patients. The intensity of the staining in intestinal epithelia was analyzed digitally and confirmed a stronger reaction in CD patients compared to UC. However, its usefulness as a diagnostic marker or its biological role in IBD requires further analyses.

### 3.3. Role of Proteomics in the Understanding IBD Pathogenesis

Zhou Z et al. [50] compared samples obtained from mucosal lesions of eight CD patients with the surrounding normal-looking mucosa. Six differentially expressed proteins were identified, including prohibitin, calreticulin, apolipoprotein A-I, intelectin-1, protein disulfide isomerase, and glutathione S transferase Pi (GSTP1). The proteins play important roles in cell differentiation and apoptosis [111,112], intracellular signaling [113], maintaining proper cell adhesion [114], and defense against microbial antigens [115], and are associated with degradation of various endogenous and exogenous metabolites [116]. The upregulation of prohibitin and GSTP1 in the inflamed mucosa was confirmed via Western blot on the small cohort of 4 other CD patients.

In 2020, Pierre N et al. [63] performed an analysis aiming at a proteomic characterization of the ileal and colonic CD (eight patients in each group). Tissue samples were taken from ulcer edges and compared with corresponding samples obtained from normal-looking mucosal regions. The authors documented a difference between ileal and colonic ulcers mainly in the expression of proteins involved in the epithelial–mesenchymal transition, neutrophil degranulation, and ribosomes. Both cohorts also showed increased expression of proteins involved in protein processing in the endoplasmic reticulum and a decrease in mitochondrial proteins. Different expressions of two proteins related to the endoplasmic reticulum stress, namely heat-shock 70 kDa protein 5 and heat-shock protein 90 kDa beta member 1, were subsequently confirmed via IHC.

Bennike TB et al. [54] compared samples obtained from non-inflamed mucosal regions in UC patients (n = 10) and HC (n = 10). Forty-six differentially expressed proteins were identified, of which eleven were associated with the formation of neutrophil extracellular traps (NET). The findings were confirmed through an increased number of neutrophils in histology and the presence of extracellular DNA in samples supporting the involvement of NET. The results suggest that innate immunity may play a role in UC pathophysiology.

Erdmann P et al. [61] correlated results from gene expression analysis and proteomics of samples obtained from inflamed and non-inflamed mucosa in 10 UC patients and from 10 controls. Several genes coding various metabolizing enzymes (i.e., cytochrome P450 family 2 subfamily C member or UDP glucuronosyltransferase family 1 member A) and transporters (ATP-binding cassette sub-family B member 1, ATP binding cassette subfamily G member 2 or monocarboxylate transporter 1) showed a decreased expression in inflamed tissue. On the other hand, the expression of multidrug resistance protein 4, organic anion transporting polypeptide 2B1, and organic cation transporter-like protein 2 were significantly elevated in the inflamed mucosa. However, on the protein level, only monocarboxylate transporter 1 was found to be significantly downregulated in both inflamed and non-inflamed mucosa of UC patients in contrast to controls.

In 2011, Zhao X et al. [47] analyzed intestinal samples from 12 patients with UC and 12 patients with HC. They identified 26 dysregulated proteins, from which a cluster of 11 proteins involved in the p38 mitogen-activated protein kinase (MAPK) pathway was defined via a subsequent Western blot analysis. An IHC analysis confirmed the upregulation of P-p38 and downregulation of MAWD binding protein and galectin in UC patients. The expression correlated with disease severity and was able to classify UC risk with high sensitivity (94.83 ± 2.91%) and specificity (98.33 ± 1.65%). The P38 MAPK pathway is suggested to be involved in patients with active UC and might be a potential biomarker for evaluating UC risk.

Poulsen NA et al. [48] focused on UC patients with active proctosigmoiditis (n = 20). Different regions of the bowel were analyzed and compared to four control patients. The authors were able to identify 43 proteins differentially expressed between inflamed and normal-appearing mucosa, which were mainly involved in energy metabolism (triosephosphate isomerase, glycerol-3-phosphate-dehydrogenase, alpha-enolase, and L-lactate dehydrogenase B-chain) and oxidative stress (superoxide dismutase, thioredoxins, and selenium binding protein).

An example of a study aiming at treatment-naïve patients was carried out by Schniers A et al. [62]. Colonic samples from the inflamed mucosa of 17 UC patients were taken and compared with a cohort of 15 HC. The downregulated proteins included metallothioneins, PPAR-inducible proteins, fibrillar collagens, and proteins involved in bile acid transport and metabolic functions of nutrients, energy, steroids, xenobiotics, and carbonate. Those proteins involved in immune response and protein processing in the endoplasmic reticulum were upregulated.

### 3.4. Proteomics and Analysis of the Disease Behavior

IBD show a substantial phenotypical variability concerning their localization, behavior, and clinical severity, which significantly affects the choice of an optimal therapeutic strategy. A basic classification of IBD to CD, UC and IBDU was not sufficient anymore and new subcategories were necessary. This situation led to the introduction of the Montreal and Paris classifications [117,118], which categorize adult and pediatric IBD patients according to their phenotypic characteristics. However, the eventual differences between the subgroups with regards to their distinctive protein profiles have not been thoroughly studied.

One of the studies addressing this issue comes from Stidham WR et al. [58]. The authors analyzed protein profiles in stricturing (n = 20) vs. inflammation-predominant (n = 20) CD to try to define biomarkers capable of discriminating between the phenotypes. Five glycoproteins were identified showing ≥20% abundance change in ≥80% of the patients with inflammatory and stricturing CD. Among the proteins, cartilage oligomeric matrix protein (COMP) and hepatocyte growth factor activator (HGFA) were elevated in the stricturing group, which was subsequently confirmed via ELISA. Patients with the stricturing CD also showed a decrease in levels of HGFA after the resection of a stenotic segment. COMP is a member of thrombospondin family and participates in extracellular matrix and tissue remodeling in response to injury [119]. It was shown that the COMP expression is induced via transforming growth factor beta (TGFβ) [120]. On the other hand, the role of HGFA in fibrogenesis is rather inhibitory. It is a serin protease produced mainly by the liver [121] and its main role is to activate the hepatocyte growth factor (HGF) in response to tissue injury [122]. HGF then serves as an antagonist to TGFβ, promoting tissue reparation and inhibiting fibrogenesis [123].

Townsend P et al. [55] performed a study examining differences in protein spectra between stricturing CD, non-stricturing CD, and UC (nine patients in each group). They identified a protein/peptide subset capable of differentiating between these three cohorts with 70% accuracy for peptides and 80% accuracy for proteins. Some of the proteins associated with the stricturing phenotype were involved in complement activation, fibrinolysis, and lymphocyte adhesion.

Appropriate monitoring of inflammatory activity is the essential prerequisite for the optimal management of IBD. It allows clinicians to decide when to make crucial treatment choices, helps prevent severe complications, and enables the achievement of a deep remission. The activity of the disease can be assessed according to clinical symptoms, endoscopic appearance, radiologic findings, or microscopy [124]. Currently, the main goal of the therapy is to achieve and maintain endoscopic remission [125,126]. However, the persisting histological activity itself is associated with an increased risk of severe complications, such as a higher hospitalization rate, a need for therapy escalation, a higher risk of surgery, or the development of neoplasia [127,128,129,130,131,132]. Histology currently represents the most delicate method of detection of persisting inflammatory activity. However, modern proteomic techniques could provide an additional tool, enabling the stratification of patients in histological remission according to their relative risk of disease recurrence or development of complications.

Kanmura S et al. [44] analyzed sera from 22 patients with CD, 48 with patients UC, 5 patients with CRC, 6 patients with infectious colitis and 13 patients with HC and searched for new biomarkers capable of proper reflecting the disease activity. They identified three proteins (human neutrophil peptide 1, 2 and 3) that displayed higher serum levels in patients with active UC compared to all other groups, including the group of patients with inactive UC. The findings were also confirmed via ELISA. The aforementioned proteins represent anti-microbial peptides and belong to the group of α-defensins. They are mainly produced by neutrophils and macrophages and could serve as biomarkers of active UC. Their levels were also significantly decreased after successful corticotherapy, suggesting their utility for the prediction of treatment outcomes.

In the study by Gruver AM et al. [68], the authors performed both targeted and global proteomic analyses of intestinal samples from 19 patients with UC. They found a positive correlation between histological scores of disease severity (namely Geboes Score and Robarts Histopathology Index) and neutrophil-associated proteins. In contrast, a negative correlation was found between the scores and cell junction proteins and β-catenin, which was probably related to the intestinal crypt disruption.

### 3.5. Proteomics for Prediction of Treatment Response

The initial treatment strategy depends on the intensity and extent of the inflammation and is further maintained or escalated according to the patient’s response. However, there is a lack of data regarding the proper identification of patients who are at increased risk of relapse or are primary non-responders to the therapy. Currently, only one-third of IBD patients achieve remission and only half of them remain relapse-free [133,134,135]. Therefore, it is necessary to identify new biomarkers that would predict treatment success and allow a more personalized therapeutic approach. Most of the proteomic studies found in this systematic search aimed at the analysis of patients treated with infliximab (IFX).

Meuwis MA et al. [41] examined serum proteomic profiles of 30 patients with CD before and after the IFX therapy to elucidate pathogenic mechanisms of primary non-responsiveness. They identified a protein platelet aggregation factor 4 that showed significantly higher levels in non-responders. The authors suggest that the response to IFX therapy may be related to platelet metabolism.

Gazouli M et al. [52] analyzed the sera of 18 CD patients grouped according to their response to the IFX therapy. The patients were divided into three groups: (1) those who achieved clinical and serological remission, (2) those who responded to therapy, and (3) primary non-responders. Several proteins mostly involved in immune responses were identified, including apolipoprotein A-I, apolipoprotein E, complement C4-B, plasminogen, serotransferrin, β-2-glycoprotein 1 and clusterin. The proteins were upregulated in both non-responders and responders, though the levels did not change in patients who achieved remission. Remitters, on the other hand, showed significantly increased levels of leucine-rich alpha-2-glycoprotein, vitamin D-binding protein, alpha-1B-glycoprotein, and complement C1r subcomponent.

Pierre N et al. [66] investigated sera of 102 patients with CD in stable clinical remission on combined antimetabolite and IFX therapy. The authors searched for potential predictors of a short-term (<6 months) and mid/long-term (>6 months) relapse. They identified 15 proteins associated with an increased risk of short-term relapse and 17 proteins associated with mid/long-term relapse. Novel biomarker combinations exhibited a high predictive capacity, as shown by their higher Z-scores than C reactive protein and fecal calprotectin levels (current references in predicting relapse).

The recent study by Liu L et al. [67] examined a cohort of 12 UC patients subdivided into responders to IFX therapy, non-responders, patients with severe UC without prior IFX therapy, and patients with mild UC without IFX therapy. Up to 257 proteins, which exhibited a different expression in responders and non-responders, were identified. After the exclusion of those caused by the severity of inflammation, the proteins were compared and verified on the gene sequence level using the Gene Expression Omnibus (GEO) database. Finally, five proteins (β-actin-like protein 2, mannose binding protein C, bactericidal permeability-increasing protein, eukaryotic translation initiation factor 3 subunit D and complement receptor type 1) were identified as potential biomarkers of non-responsiveness to IFX therapy. The proteins were subsequently confirmed in the GEO database of the microarray results from three independent cohorts of 70 human intestinal biopsies and validated using quantitative PCR data from 17 colonic mucosal biopsies.

### 3.6. Proteomics for the Selection of Biomarkers of Neoplastic Transformation

Malignant transformation is the most severe long-term complication in patients with IBD. Although IBD-associated CRC only accounts for 1–2% of patients with CRC in the general population, it constitutes 10–15% of deaths in IBD patients [136]. Among the risk factors, disease extension [137] and its duration [138] are the two major drivers. Other significant risk factors include concomitant primary sclerosing cholangitis (PSC) [139], a family history of CRC [140], and the presence of active inflammation [141]. However, the protein expression patterns of neoplastic transformation have thus far been sparsely studied.

Brentnall TA et al. [43] performed a study on 15 UC patients and 5 HC, who were divided based on the presence of dysplastic changes or cancer into progressor and non-progressor groups. Progressors were defined by the presence of high-grade dysplasia or invasive malignancy. Non-progressors were UC patients with no dysplastic changes in histology. The authors identified several proteins related to mitochondria, oxidative activity, or calcium ion binding, which were differentially expressed between the two UC cohorts. Two of the proteins, namely carbamoyl-phosphate synthase 1 (CPS1) and a S100 calcium-binding protein P, were upregulated in non-dysplastic tissue of progressors, which was confirmed via IHC on 17 additional tissue samples. S100 proteins are a large group of proteins with many functions, including maintenance the calcium homeostasis, regulation of the cytoskeleton, modification the activity of transcription factors, or regulation of protein phosphorylation [142]. CPS1 is an enzyme involved in the urea cycle and is expressed predominantly on intestinal epithelia and hepatocytes. One of its key functions is detoxifying ammonia. It also represents one of the proteins in the urea cycle responsible for the production of arginine [43]. Alterations in the urea cycle may lead to an impaired supply of arginine and subsequently affect the production of nitric oxide, which can cause DNA damage and interfere with DNA repair [143,144]. These proteins, thus, could serve as biomarkers of future progression to dysplasia. The authors also declare that the overall protein profile in the non-dysplastic tissue of progressors is closer to dysplastic tissue than to the mucosa of non-progressors, suggesting that there are early changes in the protein expression before the development of histologically visible dysplasia.

In a similar study, May D et al. [46] analyzed intestinal samples from UC patients with high-grade dysplasia or cancer (progressors) and compared them with UC patients without dysplasia (non-progressors). The samples from the progressor group were taken from both dysplastic and non-dysplastic regions. Each group included five patients. Several protein clusters were identified as displaying a different expression in dysplastic and non-dysplastic regions of progressors. The clusters included mitochondrial and cytoskeletal proteins, the RAS protein superfamily, and proteins related to apoptosis and metabolism. Two proteins, namely TNF receptor-associated protein 1 (TRAP1) and isoform 1 of carbamoyl-phosphate synthase (CPS1), were validated via IHC. The expression of TRAP1 was increased in both dysplastic and non-dysplastic tissue of progressors, while CPS1 showed a different expression between progressors and non-progressors. An IHC assessment of CPS1 could, therefore, serve as a surrogate biomarker for the prediction of neoplastic transformation.

In 2020, Merli AM et al. [65] assessed dysplastic, inflammatory, and normal-appearing mucosal regions in five patients with UC. They identified 11 proteins upregulated in dysplastic regions compared to non-dysplastic tissue. One of those proteins, i.e., a solute carrier family 12 member 2 (SLC12A2), which is a transmembrane protein important for maintaining a proper cellular ion balance and volume [145], was subsequently confirmed via IHC on the compound validation cohort including 37 UC patients with dysplasia, 14 UC patients with CRC, 23 patients with longstanding UC, 35 patients with sporadic adenomas, 57 patients with sporadic serrated lesions, 82 patients with sporadic CRCs, and mouse models. Based on the IHC assessment, the SLC12A2 could differentiate preneoplastic and neoplastic lesions from inflammatory lesions with 89% sensitivity, 95% specificity, and 92% accuracy for UC-related dysplasia.

### 3.7. Proteomics in Pediatric IBD

Proteomic studies performed on pediatric cohorts are limited in number. During this systematic search, only five publications were identified [53,57,70,87,90]. However, the need for new reliable biomarkers of diagnosis and/or disease activity is eminent, since pediatric IBD show a higher percentage of atypical phenotypes with divergent morphology and often a non-specific or even misleading clinical presentation compared to adults [2,146,147]. In addition, the significance of disease activity monitoring and its predictive value for the subsequent development of clinical complications is poorly examined in a pediatric population. This fact is especially true for microscopic inflammatory activity. In our previous studies [148,149], we showed its uncertain predictive value and poor correlation with the endoscopic findings or the clinical severity of the disease.

Vaiopoulou A et al. [53] correlated serum proteomic profiles obtained from 12 children and 12 adults with CD. Bactericidal acute-phase protein ceruloplasmin [150] and apolipoprotein B-100 were found to be increased in children with CD, while clusterin, which is a glycoprotein with anti-apoptotic activity [151], was overexpressed in adult CD patients. The overexpression was subsequently confirmed via Western blot. However, the specificity of clusterin remains unclear, since other studies [152] show its increased expression as a non-specific age-related change, suggesting no causative relation to the diagnosis of CD.

Starr AE et al. [57] focused on treatment-naïve children with IBD. In the discovery cohort of 15 children with CD, 15 with children with UC, and 20 children with HC, the authors identified 5 proteins involved in cell metabolism that are capable of differentiating between IBD and HC, as well as 12 other proteins discriminating between CD and UC. The overexpressed proteins in CD patients were involved in the metabolism of fatty acids, while those upregulated in UC were tied to energy and amino acid metabolism. Up to 116 proteins correlated with the severity of the disease. Application of the two panels to the validation cohort differentiated 95.9% of IBD patients from HC patients and 80% of CD patients from UC patients. Two proteins, namely visfatin, and metallothionein-2, were subsequently confirmed using ELISA.

In the recent study by Louis Sam Titus ASC et al. [70], the authors performed a proteomic analysis of sera from 10 children with CD, 10 children with UC, and 7 children with HC. In total, 1322 proteins were identified. Up to 129 proteins showed upregulation in the IBD group compared to HC. Seven proteins were subsequently verified via ELISA on the validation cohort of 60 children with IBD (30 CD and 30 UC) and 16 children with HC. Among the most discriminatory proteins found, resistin showed particularly high sensitivity and specificity for both CD and UC patients. The protein is highly expressed in neutrophils and macrophages and is secreted at the site of inflammation, where it attracts other immune cells [153]. It may represent a promising biomarker for pediatric IBD. Additionally, elastase and lactoferrin, as well as other proteins reflecting the increased neutrophilic activity [108,154], were consistently elevated in patients with UC, and may also offer a potential biomarker value for this condition.

### 3.8. Proteomics in PSC-IBD

Approximately 70% of patients with PSC develop intestinal inflammation in the course of time [155]. The disease is usually diagnosed as UC [156,157], albeit many authors consider it a unique form of IBD and do not follow further subclassification. Compared to conventional IBD, PSC-associated disease often manifests atypical morphology. The inflammation shows a characteristic reverse gradient with maximal disease activity in the proximal colon and a frequent rectal sparing [158]. Quite commonly, there is a discrepancy between mild endoscopic findings and severe inflammatory activity found in histology. The risk of malignant transformation is higher compared to conventional IBD. The cumulative risk of CRC development 10 or 20 years after the diagnosis is 2% and 7%, respectively, for conventional UC, and 7–14% and 18–31% for PSC-IBD [159,160,161]. The proteomic studies analyzing PSC-IBD are scarce. Our systematic search found only two original works, which are discussed below.

Corfe BM et al. [56] analyzed intestinal samples from 45 UC patients, 7 PSC-UC patients, and 10 HC. IBD patients were grouped according to disease duration, subtype, cancer risk, and inflammatory status. The groups were as follows: (1) quiescent recent-onset UC, (2) PSC-UC, (3) quiescent longstanding pancolitis, (4) active colitis and non-inflamed proximal colonic mucosa, (5) pancolitis with dysplasia (both dysplastic lesions and distal rectal mucosa), and (6) control group. The proteomic analysis focused on alterations in intermediate filaments, especially keratins. Compared to non-inflamed mucosa and HC, patients with the active disease showed a reduction in vimentin and keratins 8, 18, and 19. Upregulation of those proteins was apparent in patients with quiescent longstanding pancolitis in contrast to controls and patients with recent-onset remission. Decreased levels of several intermediate filament proteins were noted in patients with PSC-UC and UC with dysplasia. Western blot verified the implications of keratin 8. Its levels and the degree of its phosphorylation were reduced in a progressive disease and acute inflammation but appeared restored or elevated in patients with clinical and endoscopic remission. Such restoration was not seen in patients with an elevated risk of cancer. The authors conclude that the regulation of keratins in IBD in remission may be related to the subsequent cancer risk.

Vessby J et al. [69] recently assessed proteomic profiles of colonic samples from 9 patients with PSC-UC, 7 patients with UC and 7 patients with HC. All IBD patients were in clinical and histological remission without a prior episode of pancolitis. Based on their analysis, the authors selected five proteins with the highest discriminatory value between PSC-UC and UC and verified them on the validation cohort of 16 PSC-UC patients and 21 UC patients. The analysis confirmed the upregulation of protein 1-acetylglycerol-3-phosphate O-acyltransferase 1 (AGPAT1) in PSC-UC, which was subsequently confirmed by IHC. AGPAT1 is a protein responsible for the conversion of phospholipid lysophosphatic acid into phosphatic acid [162]. The results, thus, suggest a possible involvement of a lipid metabolism in the immunopathogenesis of IBD [163]. The protein is also linked to the disease progression in patients with CRC [164]. According to the authors, AGPAT1 may serve as a marker of PSC-UC in colonic samples.

## 4. Conclusions

Over the years, many potential biomarkers for IBD diagnosis, disease monitoring, therapeutic response or patients’ risk stratification were introduced; however, only a handful of them were successfully established in routine practice. Such disproportionality between the amount of proposed and validated biomarkers probably reflects both the complexity of the disease and the limitations of proteomics studies. In this review, 62 original works were reported; however, the majority of them were limited to small sample sizes, and the results lacked validation on larger cohorts. Further multicentric studies that would properly validate proposed biomarkers are necessary before their final implementation in clinical practice. Validating a biomarker requires more than just confirming it in tissue through antibody-based methods, such as IHC or Western blot. To truly confirm its usefulness, researchers should also search for the biomarker in peripheral blood or stool, enabling a non-invasive approach. However, identifying individual biomarkers in isolation may not provide a comprehensive understanding of the complex processes underlying the disease and may lack sensitivity, specificity, and predictive value. Instead, future use of panels of biomarkers that provide compound scores will likely be more effective. Last but not least, to fully understand the pathological processes involved in IBD, studies should integrate transcriptome, proteome, metabolome, and microbiome data through a multi-omic approach or even a whole interactome. However, such studies will be difficult to perform due to costly analyses, substantially reducing sample sizes. The large amount of data generated from these studies will also require a complex bioinformatic approach, and the discovery of causal relationships in this setting will not be possible without the implementation of machine learning methods. Only then will we be able to elucidate specific mechanisms of disease progression and provide optimized and patient-specific therapy.

## Figures and Tables

**Figure 1 ijms-24-09386-f001:**
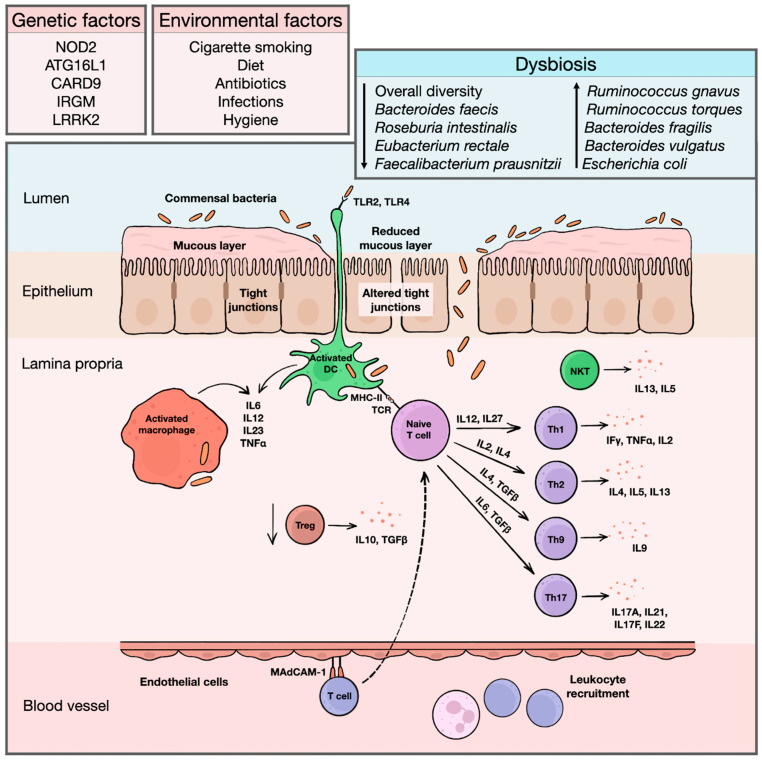
An overview of IBD etiopathogenesis. Exact etiopathogenic mechanisms of IBD initiation and progression are still largely unknown. However, the etiology of the diseases is probably multifactorial, with a contribution of both genetic and environmental factors altering physiological composition of intestinal microbiota and initiating an inadequate inflammatory response. Bacterial antigens are recognized by toll-like receptors on surface of dendritic antigen-presenting cells, macrophages and intestinal epithelia. In healthy individuals, antigen-presenting cells induce immune tolerance and stimulate differentiation of regulatory T cells, producing anti-inflammatory cytokines, such as IL10 or TGFβ. In IBD, changes in intestinal microbiota, together with increased permeability of intestinal epithelium and reduced luminal mucous layer facilitating translocation of bacterial fragments into lamina propria, triggers inflammatory response. Activated dendritic cells and macrophages produce various pro-inflammatory cytokines, such as IL2 or IL23, and initiate differentiation of naïve T cells into effector T cells. For a long time, Crohn’s disease was considered a disease driven predominantly by Th1 cells with a production of high amounts of IFNγ, TNFα, or IL2, while an ulcerative colitis was linked with Th2-predominant immune pathway and secretion of IL4, IL5, or IL13. In both diseases, there is also a strong influence of Th17-related cytokines, such as IL17A or IL17F. Apart from that, a substantial contribution of Th9 cells and natural killer T cells is recognized, further disrupting the intestinal epithelial integrity. Simultaneously, additional leukocytes are recruited from peripheral circulation. These cells bind to endothelial cells lining the mucosal microvasculature via mucosal vascular addressin–cell adhesion molecule 1, whose expression is upregulated in inflamed mucosa, and migrate to affected area of lamina propria, maintaining chronic inflammatory reaction. ATG16L1 = autophagy-related 16-like 1 gene; CARD9 = caspase recruitment domain-containing protein 9 gene; DC = dendritic cell; IBD = inflammatory bowel diseases; IFγ = interferon gamma; IL = interleukin; IRGM = immunity-related GTPase family M protein gene; LRRK2 = leucine-rich repeat kinase 2 gene; MAdCAM-1 = mucosal vascular addressin–cell adhesion molecule 1; MHC-II = major histocompatibility complex class II; NKT = natural killer T cell; NOD2 = nucleotide binding oligomerization domain containing 2 gene; TCR = T cell receptor; TGFβ = transforming growth factor beta; Th = helper T cell; TLR = toll-like receptor; TNFα = tumor necrosis factor alpha; Treg = regulatory T cell.

**Figure 2 ijms-24-09386-f002:**
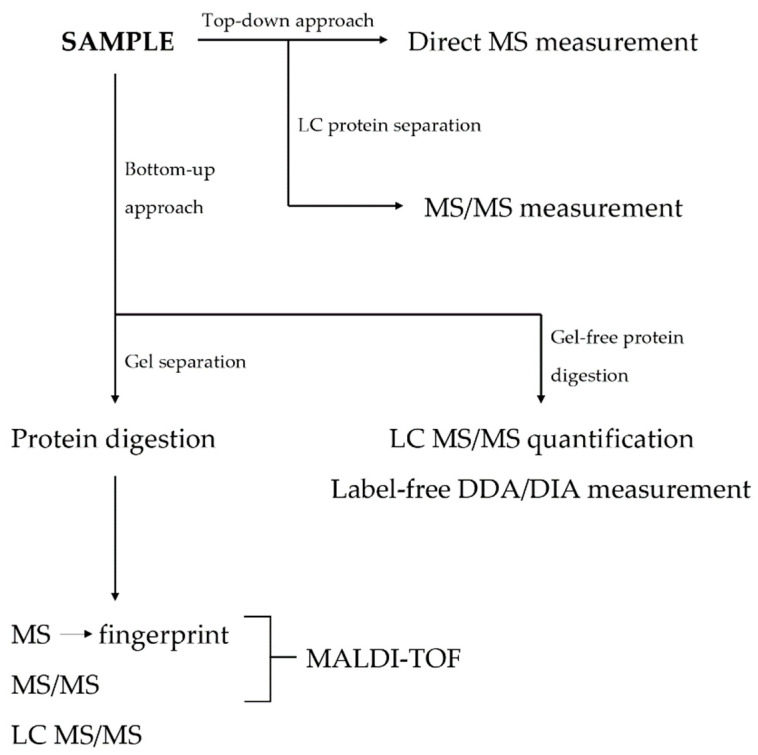
A flowchart of basic proteomic approaches. DDA = data-dependent acquisition; DIA = data-independent acquisition; LC = liquid chromatography; MALDI = matrix-assisted laser desorption/ionization; MS = mass spectrometry; MS/MS = tandem mass spectrometry; TOF = time of flight.

**Figure 3 ijms-24-09386-f003:**
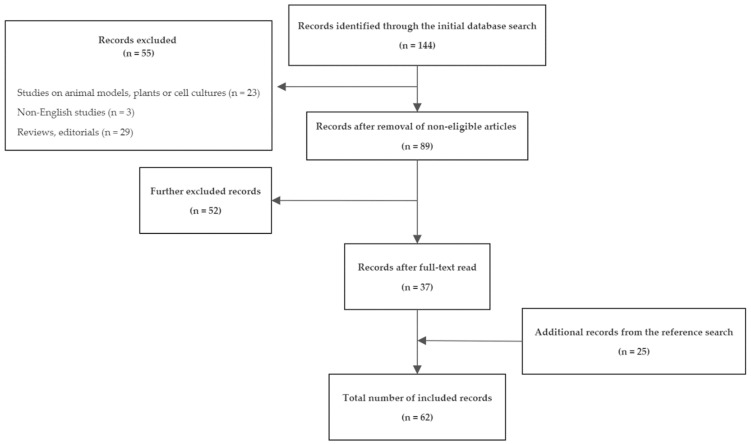
A systematic search and review flowchart.

**Table 1 ijms-24-09386-t001:** Selected proteomic studies in human IBD.

Reference	Type of the Tissue	Cohort (n)	Proteomic Technique	Main Findings
Meuwis MA et al., 2007 [40]	Serum	CD (30), UC (30), inflammatory control (30), HC (30)	SELDI-TOF-MS	Identification of four proteins (PF4, MRP8, FIBA and Hpalpha2) that could serve as potential biomarkers of IBD.
Shkoda A et al., 2007 [42]	Isolated intestinal epithelia	CD (6), UC (6), CRC (6)	2-DE and MALDI-TOF-MS	Identification of protein panels involved in signal transduction, stress response, and energy metabolism that are upregulated in IBD patients compared to controls.
Meuwis MA et al., 2008 [41]	Serum	CD (20)	SELDI-TOF-MS	Higher levels of PF4 were found in non-responders to infliximab therapy.
Brentnall TA et al., 2009 [43]	Intestinal samples	UC (15), HC (5)	iTRAQ and HPLC-TOF-MS/MS	Several proteins involved in UC neoplastic progression, including those related to mitochondria, oxidative activity, and calcium-binding, were identified. Two of them (CPS1 and S100P) were further confirmed via immuno-histochemistry.
Kanmura S et al., 2009 [44]	Serum	CD (22), UC (48), CRC (5), infectious colitis (6), HC (13)	SELDI-TOF-MS	Human neutrophil peptides 1, 2, and 3 are significantly higher in patients with active UC compared to patients with UC in remission or other diseases and decrease after successful corticosteroid therapy. The proteins may serve as biomarkers of active UC and predict treatment outcomes.
M’Koma AE et al., 2011 [45]	Intestinal samples	Colonic CD (24), UC (27)	Histology-directed MALDI-MS	Distinctive spectral peaks in submucosal layer were able to discriminate between colonic CD and UC.
May D et al., 2011 [46]	Intestinal samples	UC (15)	Label-free LTQ/Orbitrap hybrid MS coupled with nano-flow HPLC	Identification of several protein clusters differentially expressed in dysplastic and non-dysplastic mucosal regions in patients with UC-associated dysplasia or cancer. Two proteins (TRAP1 and CPS1) were detected in both dysplastic and non-dysplastic tissue via immuno-histochemistry.
Zhao X et al., 2011 [47]	Intestinal samples	UC (12), HC (12)	2-DE and MALDI-TOF-MS	Increased expression of P-p38 and decrease in MAWBP and galektin-3 were found in UC patients compared to controls and correlated with disease progression. P38 MAPK pathway is suggested to be involved in patients with active UC.
Poulsen NA et al., 2012 [48]	Intestinal samples	UC (20), HC (4)	2-DE and MALDI-TOF-MS	Forty-three proteins differentially expressed in inflamed colonic tissue in UC patients were identified. The proteins were mainly involved in energy metabolism (triosephosphate isomerase, glycerol-3-phosphate-dehydrogenase, alpha enolase and L-lactate dehydrogenase B-chain) and oxidative stress (superoxide dismutase, thioredoxins and selenium binding protein).
Seeley EH et al., 2013 [49]	Intestinal samples	Colonic CD (26), UC (26)	Histology-directed MALDI-MS	Based on 25 protein spectral peaks, a machine learning algorithm capable of differentiating between colonic CD and UC with 76.9% accuracy was constructed.
Zhou Z et al., 2013 [50]	Intestinal samples	CD (8)	2-DE and MALDI-MS	Identification of six differentially expressed proteins in mucosal lesions compared to normal intestinal mucosa, including prohibitin, calreticulin, apolipoprotein A-I, intelectin-1, protein disulfide isomerase, and glutathione S transferase Pi.
Han NY et al., 2013 [51]	Intestinal samples	CD (3), UC (4), inflammatory polyps in UC (2), HC (3)	Label-free LC/MS	Identification of three proteins (PRG2, LCP1 and PSME1) serving as potential biomarkers of active CD.
Gazouli M et al., 2013 [52]	Serum	CD (18)	2-DE and MALDI-TOF-MS	Identification of a panel of proteins (APOA1, APOE, CO4B, PLMN, TRFE, APOH and CLUS) that are upregulated in primary non-responders to infliximab therapy.
Vaiopoulou A et al., 2015 [53]	Serum	CD (12), pCD (12)	2-DE and MALDI-TOF-MS	Ceruloplasmin and apolipoprotein B-100 are increased in children with CD, while clusterin is overexpressed in adult CD patients.
Bennike TB et al., 2015 [54]	Intestinal samples	UC (10), HC (10)	Label-free LC/MS	Identification of 46 proteins differentially expressed in UC compared to controls. The proteins were often associated with neutrophils and neutrophil extracellular traps formation, suggesting involvement of the innate immunity.
Townsend P et al., 2015 [55]	Serum	Stricturing CD (9), non-stricturing CD (9), UC (9)	LC-MS	Identification of the peptide/protein subset discriminating between the three cohorts with 70% accuracy for peptides and up to 80% for proteins. Several proteins distinguishing the stricturing CD were involved in complement activation, fibrinolysis, and lymphocyte adhesion.
Corfe BM et al., 2015 [56]	Intestinal samples	UC (45), PSC-UC (7), HC (10)	iTRAQ and HPLC-ESI/Q-TOF-MS/MS	Downregulation of keratins 8, 18, and 19 and vimentin in patients with acute distal UC compared to controls and samples from non-inflamed proximal mucosa. Upregulation of those proteins in patients with quiescent longstanding pancolitis in contrast to controls and patients with recent-onset remission. Decreased levels of several intermediate filament proteins in patients with PSC-UC and UC with dysplasia.
Starr AE et al., 2017 [57]	Intestinal samples	Discovery cohort: pCD (15), pUC (15), pHC (20) Validation cohort: pCD (15), pUC (15), pHC (19)	SILAC and HPLC-ESI-MS/MS	Two protein panels were identified that discriminate IBD from HC (FABP5, NAMPT, UGHD, LRPPRC and PPA1) and CD from UC (HADHB, SEC61A1, SND1, LAP3, LTA4H, MT2A, SLC25A1, HNRNP H3, TF, ECH1, TFRC and B2M), respectively. Application of the two panels to validation cohort differentiated 95,9% IBD patients from HC and 80% CD from UC. One hundred and sixteen proteins correlated with the severity of the disease. Visfatin and metallothionein-2 were subsequently confirmed via ELISA on an independent cohort.
Stidham WR et al., 2017 [58]	Serum	CD (40)	LC-MS	Identification of five glycoproteins showing ≥20% abundance change in ≥80% of the patients with inflammatory and fibrostenotic phenotype of CD. Among the glycoproteins, COMP and HGFA were elevated in the fibrostenotic group.
Moriggi M et al., 2017 [59]	Intestinal samples and isolated intestinal epithelia	CD (30), UC (30), controls (16)	2-DE and MALDI-TOF-MS	Identification of proteins increased or decreased in inflamed and non-inflamed IBD patients, reflecting different patterns of extracellular matrix and cytoskeleton rearrangement and changes in cellular metabolism and autophagy.
Ning L et al., 2019 [60]	Intestinal samples	Discovery cohort: CD (9), UC (9), HC (6) Validation cohort: CD (3), UC (3), HC (3)	LC-MS/MS	Identification of protein spectra differentially expressed in CD, UC, and controls. Several novel proteins, including CD38, were introduced. CD38 expression was higher in IBD patients compared to controls, and higher in CD patients in comparison to UC. It was also more abundant in inflamed regions of the bowel.
Erdmann P et al., 2019 [61]	Intestinal samples	UC (10), HC (10)	LC-MS/MS	Identification of several genes for metabolizing enzymes (i.e., CYP2C9 or UGT1A1) and transporters (ABCB1, ABCG2 or MCT1) decreased during the inflammation. On the other hand, MRP4, OATP2B1, or ORCTL2 were significantly elevated in inflamed tissue. On the protein level, these results could be confirmed only for MCT1.
Schniers A et al., 2019 [62]	Intestinal samples	UC (17), HC (15)	LC-MS/MS	Downregulation of several proteins, including metallothioneins, PPAR-inducible proteins, fibrillar collagens, and proteins involved in bile acid transport and metabolic functions of nutrients, energy, steroids, xenobiotics, and carbonate, in patients with UC. Proteins involved in immune response and protein processing in endoplasmic reticulum were upregulated.
Pierre N et al., 2020 [63]	Intestinal samples	CD (16)	Label-free UPLC-ESI-MS/MS	Different proteins were expressed in ulcer edges from patients with ileal and colonic CD. The proteins were mainly associated with epithelial-mesenchymal transition, neutrophil degranulation, and ribosomes.
Arafah K et al., 2020 [64]	Intestinal samples	Discovery cohort: CD (9), UC (9); Validation cohort: CD (62), UC (51)	LC-MS/MS	Identification of several proteins upregulated in CD patients, including proteins related to neutrophilic activity and damage-associated molecular patterns. Aldo-keto reductase family 1 member C3 protein was found in 8/9 CD patients and no UC patients.
Merli AM et al., 2020 [65]	Intestinal samples	Discovery cohort: UC (5); Validation cohort: UC (74), sporadic lesions (174), HC (18)	Label-free LC-MS/MS	Proteomic analysis of dysplastic, inflammatory, and normal mucosal regions in UC patients. Eleven proteins were found to be more abundant in dysplastic foci, including solute carrier family 12 member 2, which was subsequently confirmedvia immuno-histochemistry.
Pierre N et al., 2020 [66]	Serum	CD (102)	UPLC-MS	Identification of several circulating biomarkers associated with risk of short-term and mid/long-term relapse after infliximab withdrawal.
Liu L et al., 2022 [67]	Intestinal samples	UC (12)	Label-free LC-MS	Identification of five proteins (ACTBL2, MBL2, BIP, EIF3D, and CR1) as potential predictive biomarkers of non-response to infliximab therapy.
Gruver AM et al., 2022 [68]	Intestinal samples	UC (19)	MS	Neutrophil-related proteins correlated with histological scoring indices of disease severity. A negative correlation was found between disease severity and cell junction proteins and β-catenin.
Vessby J et al., 2022 [69]	Intestinal samples	Discovery cohort: PSC-UC (9), UC (7), HC (7) Validation cohort: (PSC-UC (16), UC (21)	LC-MS/MS	1-acetylglycerol-3-phosphate O-acyltransferase 1 was proved to be higher in PSC-UC patients compared to UC. The finding was confirmed via immuno-histochemistry.
Louis Sam Titus ASC et al., 2022 [70]	Serum	Discovery cohort: pUC (10), pCD (10), pHC (7) Validation cohort: pCD (30), pUC (30), pHC (16)	Aptamer-based targeted proteomic assay	Significant elevation of serum resistin, elastase, and lactoferrin in both pCD and pUC patients. The proteins (especially resistin) may serve as serum biomarkers of pIBD.

2-DE = two-dimensional gel electrophoresis; CD = Crohn’s disease; CRC = colorectal cancer; ESI/Q = electrospray ionization quadrupole; HC = healthy control; HPLC = high-performance liquid chromatography; IBD = inflammatory bowel disease; LC = liquid chromatography; MALDI = matrix assisted laser desorption/ionization; MS = mass spectrometry; pCD = pediatric Crohn’s disease; pHC = pediatric healthy control; PSC = primary sclerosing cholangitis; pUC = pediatric ulcerative colitis; SELDI = surface-enhanced laser desorption/ionization; SILAC = stable isotope labeling by amino acids in cell culture; TOF = time of flight; UC = ulcerative colitis; UPLC = ultra-high-performance liquid chromatography.

**Table 2 ijms-24-09386-t002:** A list of candidate biomarkers and their potential clinical applications.

Candidate Biomarkers	Potential Utility
Bone marrow proteoglycan, calprotectin, carbonyl reductase, **CD38**, chitinase 3-like 1, collagen type VI alpha 1 chain, detyrosinated α-tubulin, FIBA, fibrinogen, focal adhesion kinase, GDP-forming subunit and periostin, hpalpha2, intelectin 1, keratin 19, L-lactate dehydrogenase, L-plastin, MRP8, myeloperoxidase, neutrophil defensin-1, olfactomedin 4, PF4, proteasome activator subunit 1, RHO-associated protein kinase 1, Rho-GDP, spectrin repeat containing nuclear envelope 2, succinate CoA ligase 2, tau tubulin kinase 2, tenascin C, vimentin, and vinculin.	Distinguishing IBD from other diseases
**Aldo-keto reductase family 1 member C3 protein**, cathepsin G, Ig mu chain C region protein group, Ig mu heavy chain disease protein, keratin 4, lactotransferrin, lysozyme C, neutrophil elastase, and proteins S100 A8 and A9.	Differentiating CD from UC
Apolipoprotein A-I, calreticulin, disulfide isomerase, glutathione S transferase Pi, **heat-shock 70 kDa protein 5, hHeat-shock protein 90 kDa beta member 1**, intelectin-1, and prohibitin.	Pathogenesis of CD
α-enolase, fibrillar collagens, **galectin**, glycerol-3-phosphate-dehydrogenase, L-lactate dehydrogenase B-chain, **MAWD binding protein**, metallothioneins, monocarboxylate transporter 1, **P-p38**, PPAR-inducible proteins, selenium binding protein, superoxide dismutase, thioredoxins, and triosephosphate isomerase. Spectrum of proteins associated with neutrophil extracellular traps and involved in bile acid transport and metabolic functions of nutrients, energy, steroids, xenobiotics, and carbonate.	Pathogenesis of UC
**Cartilage oligomeric matrix protein** and **hepatocyte growth factor activator**.	Stricturing phenotype of CD
β-catenin, cell junction proteins, **human neutrophil peptides 1, 2, and 3**, and neutrophil-associated proteins.	Assessment of the inflammatory activity in UC
α-1B-glycoprotein, apolipoprotein A-I, apolipoprotein E, β-actin-like protein 2, β-2-glycoprotein 1, bactericidal permeability-increasing protein, clusterin, complement C1r subcomponent, complement C4-B, complement receptor type 1, eukaryotic translation initiation factor 3 subunit D, leucine-rich alpha-2-glycoprotein, mannose binding protein C, plasminogen, protein platelet aggregation factor 4, serotransferrin, and vitamin D-binding protein.	Response to infliximab therapy
**Carbamoyl-phosphate synthase 1, S100 calcium-binding protein P, solute carrier family 12 member 2, TNF receptor-associated protein 1, and isoform 1**.	Neoplastic transformation in UC
Apolipoprotein B-100, **ceruloplasmin**, **clusterin**, elastase, lactoferrin, **metallothionein-2**, resistin, and **visfatin**.	Distinguishing pediatric IBD from adult IBD and other diseases
**1-acetylglycerol-3-phosphate O-acyltransferase 1**, keratin 8, 18, and 19, and vimentin.	Pathogenesis of PSC-IBD (keratin 8 also involved in neoplastic transformation)

Candidate biomarkers subsequently confirmed via immunochemical methods are highlighted in bold. CD = Crohn’s disease; IBD = inflammatory bowel diseases; PSC = primary sclerosing cholangitis; UC = ulcerative colitis.

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
