# Peer review of "A Current State of Proteomics in Adult and Pediatric Inflammatory Bowel Diseases: A Systematic Search and Review"

_ijms, 2023, doi:10.3390/ijms24119386_

Round 1

Reviewer 1 Report

Proteomics, the study of a wide range of proteins in tissues, has the potential to bridge the gap between the genome, transcriptome, and phenotypic expression of the disease. By analyzing proteins, proteomics offers a promising approach to identify new biomarkers. This systematic search and review summarize the current progress of proteomics in human IBD. It explores the utility of proteomics in research, describes the fundamental techniques used in proteomic analysis, and provides an up-to-date overview of existing studies conducted in both adult and pediatric IBD populations.

Authors have done a good job for combined IBD analysis but it would have made things much clear if the IBD had been split into ulcerative colitis (UC) and Crohn’s disease (CD) or separate heading for adults an dpediatrics. Technique or source of proteome based study could also have benefitted.

Two tables have been added, one showing selected studies, other detailed in supplementary. Criteria for selective are not defined. Despite attaching PRISMA document, authors have done some haphazard things. They need to do more in coherence. When many authors are working, this is a common happening. I would suggest them to all go through and raise points to make it coherent so all can follow what is being conveyed and how it should be done.

No discussion section and hence, limitations of this study or future prospects missing (PRISMA guideline section 23 a, b, c, d.)

Biomarker search is the major aim through proteomics but I do not see any specific heading or table on that. Just few scattered sentences on that.

Another figure and table could benefit.

While the findings of the study contribute to the broader field of IBD research and patient care, narrowing it down at some points could be beneficial. Address the potential impact of proteomics on understanding disease mechanisms, improving diagnosis, guiding treatment decisions, and monitoring disease progression for both UC and CD.

Should be edited at some places.

Reviewer 2 Report

This review article gives an update on the current state of research on proteomics in inflammatory bowel diseases (IBD). The author systematically searched and reviewed available studies in both adult and pediatric IBD, discussing the usefulness of proteomics in research, explaining basic proteomic techniques, and presenting an up-to-date summary of the studies. The review is well-written and informative, covering the need for new biomarkers and the potential of proteomics to meet that need. The author also explains why proteomics is a promising alternative to genomic and transcriptomics studies and the limitations of understanding the pathological processes involved in IBD. Although the article is useful for researchers and clinicians in the field of IBD, some minor areas need improvement, which will be discussed below.

1.       Section 1.3: one potential improvement could be to provide more detail on the advantages and disadvantages of ionization sources used in the IBD field. For example, the section briefly mentions MALDI, ESI and FTIC, and TOF, but it could benefit from elaborating on these limitations and how they compare to the advantages of other techniques. Especially, the author used MALDI as one of the main keywords in the search criteria, missing the reason why this technique is preferable one compared to other ionization sources in the IBD field. The author included the DDA and DIA measurements in Figure 2, but DDA and DIA techniques are not mentioned in the main text at all. The author could elaborate on the advantages and disadvantages of DDA and DIA and explain why these techniques are preferred in gel-free samples.

2.       Section 2 Methods: there are a few minor areas that could be improved to enhance the clarity and transparency of the study.   

a.       The dates of the search should be explicitly stated in the review. While it is mentioned that studies published between January 2000 and January 2023 were included, it is not clear when the search was conducted. This information is important for readers to understand the currency of the search and any potential studies that may have been missed.

b.       The choice of keywords used in the search strategy is appropriate, but it may have been useful to include additional synonyms for IBD, such as “colitis” and “bowel inflammation”. While these terms are encompassed in the search string, their inclusion could have increased the sensitivity of the search.

3.       Section 3 Results and Discussion

a.       The potential findings of biomarkers in each subsection should be presented in a tabular format to improve readability and facilitate comparison within the section based on their findings instead of a big table in the end and sorted based on reference. For example, biomarkers A, B, and C in subsection 3.1 are potential biomarkers to distinguish IBD patients from other intestinal diseases or healthy controls. X, Y, and Z in section 3.3 are potential biomarkers found in IBD pathogenesis, and on. This can help readers find potential biomarkers easier based on their interest in the topic.

b.       Finally, it would be useful to provide a brief overview of the statistical analysis methods used in each proteomics study.

4.       A typo in “1.2. The value of proteomics”.

Reviewer 3 Report

Fabian et al. is a systematic search and review done in accordance with the PRISMA reporting guidelines summarizing the current state of proteomics in human inflammatory bowel disease (IBD). It comments on the utility of proteomics in research, describes the basic proteomic techniques and provides an up-to-date overview of available studies, comment on the value of proteomics in biomarker identification and provide a list of available studies in both adult and pediatric IBD. The overall search and summary of the paper is good. Sixty two original works were herein reported. True, evaluating protein patterns in specimens has potential for diagnostic and prognostic medicine by identifying integrally independent, phenotype-specific cellular and molecular characteristics. Techniques in proteomics such as mass spectrometry (MS) and imaging (I) MS are analytical technologies that directly measure molecular species in clinical specimens, contributing to the in-depth understanding of biological molecules. The biometric-system complexity and functional diversity is well suited to proteomic and diagnostic studies. The direct analysis of cells and tissues by Matrix-Assisted-Laser Desorption/Ionization (MALDI) MS/IMS has relevant medical diagnostic potential. MALDI-MS/IMS detection generates molecular signatures obtained from specific cell types within tissue sections. Herein, Dr. Ondrej Fabian and his colleagues discussed a perspective on the use of MALDI-MS/IMS and bioinformatics technologies for detection of molecular-biometric patterns and identification of differentiating proteins. They also discuss a perspective on the global challenge of transferring technologies to clinical laboratories dealing with IBD issues. The significance of serologic-immunometric advances in reference to Dr. Maria Gazouli (49, 50) is also discussed making the review interesting.

This is a well summarized, fundamental package of review which, I think is of educational values and informative in the field of molecular sciences, especially in IBD diagnostics and prognostics medicine.

Round 2

Reviewer 1 Report

Acceptable in present form